# The Dynamics of Vegetation Structure, Composition and Carbon Stock in Peatland Ecosystem of Old Secondary Forest in Riau and South Sumatra Provinces



I Wayan Susi Dharmawan [1,*], Nur M. Heriyanto [1], Raden Garsetiasih [2], Rozza Tri Kwatrina [1], Reny Sawitri [1], Denny [1], Titiek Setyawati [1], Pratiwi [1], Budi Hadi Narendra [1], Chairil Anwar Siregar [1] and Ilham Kurnia Abywijaya [1]

1    Research Center for Ecology and Ethnobiology, National Research and Innovation Agency (BRIN), Jalan Raya Jakarta-Bogor Km 46, Cibinong 16911, Indonesia; nurm012@brin.go.id (N.M.H.); rozz001@brin.go.id (R.T.K.); reny004@brin.go.id (R.S.); denn006@brin.go.id (D.); titi025@brin.go.id (T.S.); prat007@brin.go.id (P.); budi065@brin.go.id (B.H.N.); chai008@brin.go.id (C.A.S.); ilha006@brin.go.id (I.K.A.)
2    Research Center for Applied Zoology, National Research and Innovation Agency (BRIN), Jalan Raya Jakarta-Bogor Km 46, Cibinong 16911, Indonesia; rade040@brin.go.id
*    Correspondence: iway028@brin.go.id

**Abstract:** Lowland tropical rainforests provide an abundance of biodiversity as well as dynamic and stable ecosystems. These forests include tropical peat forests in various locations and forest types that have vegetation structure and composition characteristics, and carbon stocks that still need to be explored more deeply. Research on the structure and the composition of the vegetation and carbon stock in the old secondary peat forest was carried out in the protected areas of Bukit Batu, Riau Province, and Muara Merang, South Sumatra Province. Based on a 1-hectare permanent plot established in Bukit Batu and Muara Merang, 25 subplots of 20 m by 20 m were established in each location for measurement purposes. The results showed that Bukit Batu and Muara Merang had 42 and 36 species belonging to 26 and 20 families, respectively. Bukit Batu had a species diversity index (H′) of 2.93, and the dominant tree species were *Palaquium xanthochymum* with an importance value index (IVI) = 66.27%, *Eugenia* sp. (IVI = 32.76%), and *Litsea* sp. (IVI = 18.39%). The Muara Merang location had a species diversity index (H′) of 2.82, and the dominant tree species were *Eugenia* sp. (IVI = 60.88%), *Alseodaphne insignis* (IVI = 26.34%), and *Adenanthera pavonina* (IVI = 22.11%). In Bukit Batu, forest stands with a diameter of ≥10 cm contained a biomass of 178.10 tonnes/ha and carbon stock of 83.70 tonnes C/ha, which is equal to 307.20 tonnes $CO_2$/ha. Meanwhile, in Muara Merang, it was 190.41 tonnes/ha and 89.49 tonnes C/ha, which is equal to 328.44 tonnes $CO_2$/ha. This research, especially that in Bukit Batu, Riau Province, enriches the data and information available to date and is very useful in supporting restoration practices in Riau Province's Giam Siak Kecil Biosphere Reserve, which was designated by UNESCO as part of the Man and Biosphere Program.

**Keywords:** vegetation composition; carbon stock; old secondary forest; restoration; biomass



## 1. Introduction

Indonesia is an archipelagic country with tropical rainforests of around 120.5 million hectares or 63% of its total land area [1]. Tropical rainforests are rich in species and provide a stable environment [2]; they are essential for the regulation of the climate, water, and carbon cycles, as well as the protection of biodiversity on the land [3]. However, due to deforestation, the functions of tropical forests are being disrupted. Diaz et al. [4] pointed out that deforestation has various consequences, including habitat degradation and biodiversity loss, a decrease in water quality and quantity, air pollution, and an increase in greenhouse gas emissions, which contribute to climate change.

Peatland is one of the most important ecosystems in the world. Globally, about 3% of the planet's surface is covered by peatland; it is spread across temperate, boreal, subtropical,

and tropical zones and plays a crucial role in maintaining the global ecosystem balance, storing carbon, reducing greenhouse gas emissions, protecting biodiversity, and supporting human life [5]. In fact, most peatlands exist as peat swamp forest landscapes and are perceived as fragile ecosystems. Any disturbances to the vegetation can have an adverse effect on the peat's characteristics and its hydrological function. However, the peat swamp forest ecosystem is currently facing threats due to the exploitation of forests and the use of forest land for cultivation [6]. This ecosystem stores between one-third and one-half of the world's soil carbon pool, approximately 88.6 Pg-C [7]; thus, it plays a significant role in moderating the global climate. Indonesia has a massive area of tropical peatland, covering about 14.9 Mha [8,9]. Recently, Anda et al. [10] presented new, detailed spatial information on peatlands and showed that they cover 13.43 million ha, distributed across four islands as follows (in million ha): Sumatera (5.85), Kalimantan (4.54), Papua (3.01), and Sulawesi (0.024). These peatlands are globally recognized as one of the richest ecosystems in terms of carbon (C). In relation to the current global climate change issue, the C pool in the peatlands of Indonesia is of great significance as they accumulate and conserve as much as 17–19% (65 Gt) of the global peat C pool [7].

Lowland tropical rainforests, including peat swamp forests, typically exhibit high biodiversity and relatively high biomass [11]. However, deforestation and frequent fires can diminish both biodiversity and biomass. Between 1990 and 2010, factors that contributed to their endangered status included habitat loss, degradation, fragmentation, and the overexploitation of timber and other resources. Therefore, conservation efforts are crucial to prevent their extinction. Southeast Asia's peat swamp forests, which cover an area of almost 5.4 Mha, experienced an annual deforestation rate of 3.7% [12]. In Indonesia, the annual total emissions fluctuate significantly, mostly due to unpredictable peatland megafires [13,14]. From 2000 to 2016, carbon dioxide ($CO_2$) emissions from the forestry sector (logging and fires) averaged 0.71 Gt $CO_2$-e [15,16]. According to Carbon Brief [17], emissions from land use, land use change, and forestry (LULUCF) accounted for 2.4 billion tonnes of $CO_2$, which was the equivalent of Indonesia's total greenhouse gas emissions in 2015, when it was hit by the El Nino phenomenon.

Indonesia's extensive tropical forests often experience forest fires during the dry season [18,19]. Following the 2015 peatland fire in Indonesia, limited vegetation that was suitable for seedlings and other plants was observed, except for a small number of fern species that exhibited the highest density [20]. However, in tropical regions, tree growth is faster than in subtropical areas, making peat forests crucial for the absorption of greenhouse gas emissions that contribute to unwanted climate changes and vulnerability [21]. The landscape is managed by *Kawasan Hidrologi Gambut* (KHG) or the Peat Hydrology Unit [22].

The existing peat forests play a significant role in maintaining biodiversity and the carbon stock stored in the ecosystem. These areas represent the diversity of the remaining secondary peat forest vegetation in tropical regions. This is important because there is currently a dearth of field data on C dynamics, including stocks, emissions, and sequestration in tropical peatlands [23,24]. Studies of the carbon stock conditions of several peat forests in Indonesia showed a carbon stock of 73.08 tons C/ha in burned forests with diameters ≥10 cm in Central Kalimantan's peat forests [25]; in Kalimantan forests, the carbon sequestration from biomass, which is influenced by tree diameter, leads to an annual increase of 2.7 ± 0.5 Mg-C ha−1 in peat swamp [24]. Miettinen et al. [26] stated that Sumatra Island has been left with just 28% of its historical forested peatlands; thus, these data and valuable insights are important in supporting restoration efforts and spatial planning around the management area. Studies showed that the carbon stock of degraded peat swamp forests in the Meranti Islands, Riau, Sumatra, is 39.47 tonnes C/ha for the tree levels [27], while in the Bengkalis' peat swamp forests in Sumatra, carbon stocks were recorded as 151.14 tonnes C/ha in old secondary forests, 43.42 tonnes C/ha in young secondary forests, and 36.37 tonnes C/ha in old shrublands [28]. However, data on the vegetation structure and composition and the carbon stocks of secondary peat forests, with a specific focus on protected areas such as conservation forests or biosphere reserve areas, are still limited.

This study aims to assess vegetation structure, composition, and carbon stock potential in protected areas by focusing on secondary peat forests in Sumatra, Indonesia. By addressing the knowledge gaps, we aim to support conservation and sustainable management efforts and contribute to biodiversity preservation and climate change mitigation at the regional and global levels.

## 2. Materials and Methods

### 2.1. Research Location

This research was conducted in two locations (Figure 1). These locations are categorized as old secondary forests in protected areas. Location 1 (designated by UNESCO's Man and Biosphere Program) in Bukit Batu, Riau Province, is situated at the coordinates 1°18′02.2″ N; 101°57′53.10″ E and is administratively located within Api-Api Village, Bandar Laksamana District, Bengkalis Regency, Riau Province. Location 2 (Muara Merang) is located at 1°52′54.7″ S; 104°07′45.7″ E and is administratively located within Muara Merang Village, Bayunglincir District, Musi Banyuasin Regency, South Sumatra Province.

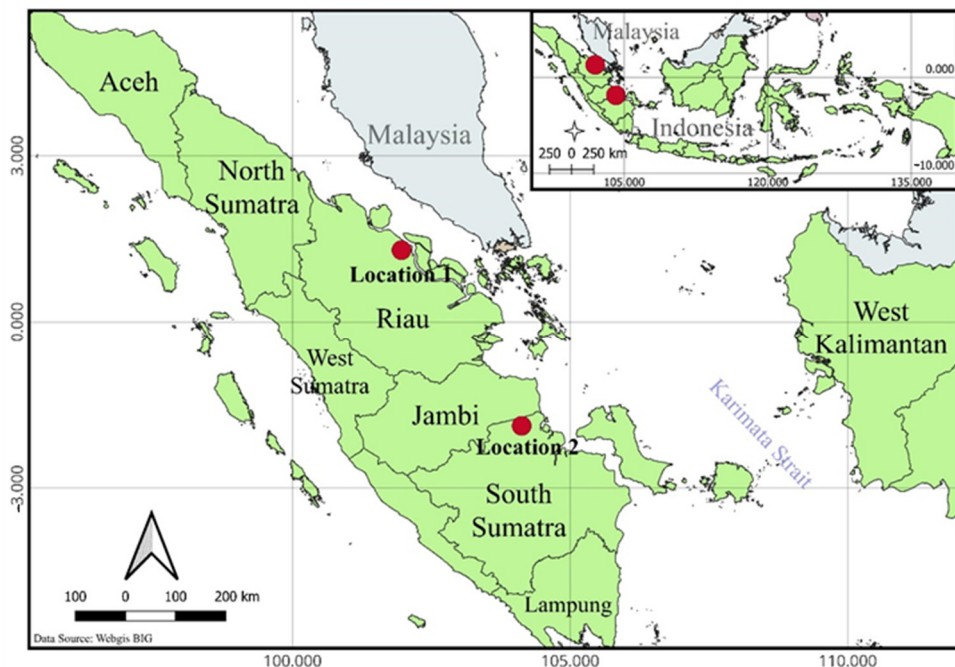

**Figure 1.** Research locations.

Both locations have elevations ranging from 20 to 22 m above sea level, and both constitute part of the tropical peat swamp rainforest. The terrain is flat, with 0–8% gradients. The soil in the research locations comprises Tropochemicals, Troposaprists, and Tropofibrist Sapric, which have undergone advanced decomposition and originate from old alluvial sediments. The rock constituents include clay, silt, gravel, plant residues, and sand, with colors ranging from dark brown to black, and when squeezed, the fiber content is less than 15% [29–32].

According to the classification by Schmidt and Ferguson, the climate in these areas (locations 1 and 2) falls under type A, with 2890 and 2958 mm of annual rainfall on average, as well as 208 and 207 rainy days on average. The highest monthly rainfall in locations 1 and 2 occurs in March (146 mm) and February (158 mm), respectively, whereas the lowest is 38.5 and 36.5 mm, respectively, and occurs in June [33,34].

### 2.2. Experimental Design and Sampling

The stand conditions in each research location were relatively uniform; therefore, an area of one hectare (100 m × 100 m) was sampled to represent the stands in each location.

The research study was conducted from May 2022 to August 2022. In a 1-hectare plot, there were 25 nested subplots of 20 m × 20 m for tree measurements, 5 m × 5 m belt transects, and 2 m × 2 m areas for seedling measurements (Figure 2).

All species names of the trees within the belt transect were recorded, and the heights and stem diameters of the trees were measured; the species names of the seedlings were recorded and quantified. Samples of unidentified materials were collected for identification at the Forest Research and Development Laboratory, Bogor. The World Flora Online database [35] was referred to for species nomenclature.

In each location, a 100 m × 100 m (1 hectare) plot was established, which was then subdivided into 20 m × 20 m subplots, resulting in a total of 25 subplots within one plot (Figure 2). The tree, belt, and seedling levels were determined according to the following criteria [28,36]: (1) trees with a diameter ≥10 cm at a height of 1.3 m; if buttressed, the diameter was measured 20 cm above the buttress; (2) belts with a diameter <10 cm and height exceeding 1.5 m; and (3) seedlings with a height of less than 1.5 m, including sprouts.

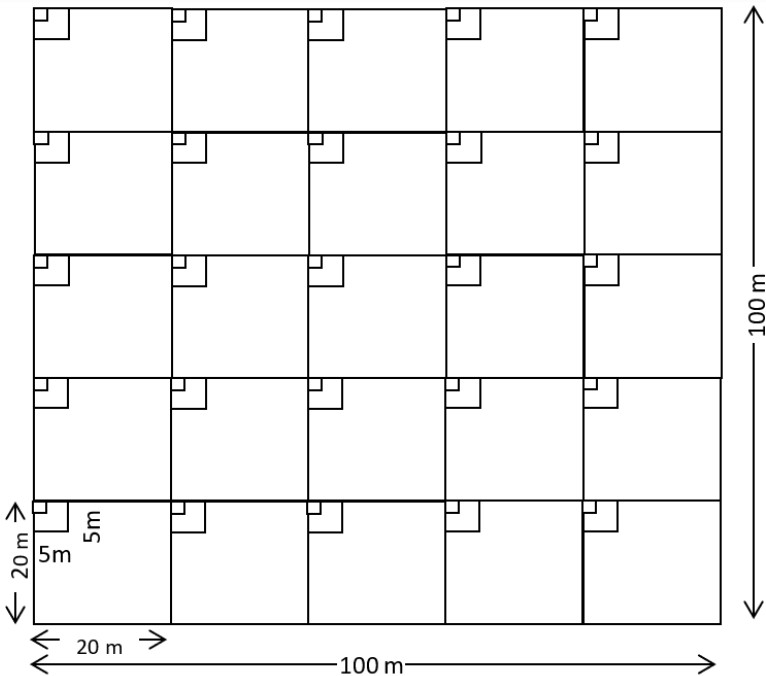

**Figure 2.** Research measurement plot in Bukit Batu and Muara Merang.

*2.3. Data Analysis*

All scientific names of the plants were standardized according to the accepted name from the World Flora Online database [35] using the R package "WorlFlora" [37]. To investigate the species richness and phylogenetic diversity of the plants, phylogenetic dendrograms (phylograms) were generated and visualized using the R package "V.PhyloMaker2" [38] and "phytools" [39], respectively. The tree endemism status was verified using the BGCI Global Tree Portal database [40].

The analysis of the dominant species was carried out through the calculation of the importance value index (IVI). The IVI, an ecological term, provides a measurement (as a percentage) of how dominant a species is in an ecosystem; the higher the IVI value of a species, the more dominant the species [41–43]. The IVI was calculated using the following parameters: relative dominance [44–46]. The IVI is obtained by adding up each species' relative density, relative frequency, and relative dominance, using the following formula (Equations (1)–(3)):

$$\text{Relative density}(\%) = \frac{\text{number of individuals for a species}}{\text{total number of individuals for all species in the plot}} \times 100\% \quad (1)$$

$$\text{Relative frequency}(\%) = \frac{\text{frequency of a species}}{\text{total number of frequency for all species in the plot}} \times 100\% \tag{2}$$

$$\text{Relative dominance}(\%) = \frac{\text{total basal area of a species}}{\text{total basal area of all species in the plot}} \times 100\% \tag{3}$$

To calculate the stand's species diversity index, which is also known as the Shannon index, the Misra [47] formula was used (Equation (4)).

$$\text{H}' = -\sum_{i=1}^{n}\left(\frac{\text{ni}}{\text{N}}\right)^2 \text{Log e}\left(\frac{\text{ni}}{\text{N}}\right) \tag{4}$$

where H′ is the Shannon index, ni is the importance value of each species, e is the constant, and N is the total importance value.

There are three growth stages that correspond to the potential of the plant species in the research plot, namely seedling, sapling, and tree, each of which is calculated per unit area (ha). In addition to the stand volume, the stand potential was computed. The number of stems per hectare was categorized into five diameter classes: 10 to 19 cm, 20 to 29 cm, 30 to 39 cm, 40 to 49 cm, and ≥50 cm. In this research, the above-ground biomass was measured using Chave's formula (Equation (5)); thus, it was not necessary to employ a destructive sampling approach [48].

$$Y = 0.0673 \times (\rho D^2 H)^{0.976} \tag{5}$$

where the variables Y, D, ρ, and H represent the total biomass (kg), diameter at breast height (cm), wood density (gr/cm$^3$), and height (m), respectively. The wood density (ρ) values were obtained from references, according to the species found [49,50].

The carbon stock in plants and the carbon dioxide sequestration were calculated using Equations (6) and (7), respectively [51,52]:

$$\text{Carbon stock} = \text{Dry weight of plant} \times 47\% \tag{6}$$

$$\text{Carbon dioxide sequestration (CO}_2) = 44/12 \times \text{carbon stock} \tag{7}$$

The use of these formulas depends on the climate conditions in the study site; the study site in this case has an annual rainfall of 2141 mm/year and falls under the moist category (rainfall at a rate of 1500–4000 mm annually). Microsoft Excel Windows 10 software was used to tabulate and analyze the acquired data [53].

**3. Results**

*3.1. Composition and Vegetation Potential*

3.1.1. Species Composition

There were 67 species observed (belonging to 49 genera, 33 families) in the two locations combined; 42 species (34 genera, 26 families) were recorded in Bukit Batu, and 36 species (28 genera, 20 families) were recorded in Muara Merang, as shown in Figure 3.

The plant families with the highest number of species found across all of the studied locations were Lauraceae and Dipterocarpaceae. In the Muara Merang forest, 39 tree species were found, classified into 20 families; the most abundant species belonged to Lauraceae, Dipterocarpaceae, and Anacardiaceae. In this study, Bukit Batu had 38 tree species with diameters ≥10 cm, totaling 528 stems/ha across the 25 subplots of 20 m by 20 m, while Muara Merang had 565 stems/ha.

The research results in Bukit Batu and Muara Merang showed that there were nine dominant species with an important value index (IVI) >10%. The stand diversity index (H′) of 2.93 and 2.82 fell into the moderate category [54]. Table 1 displays the density and important value indexes for these dominant tree species.

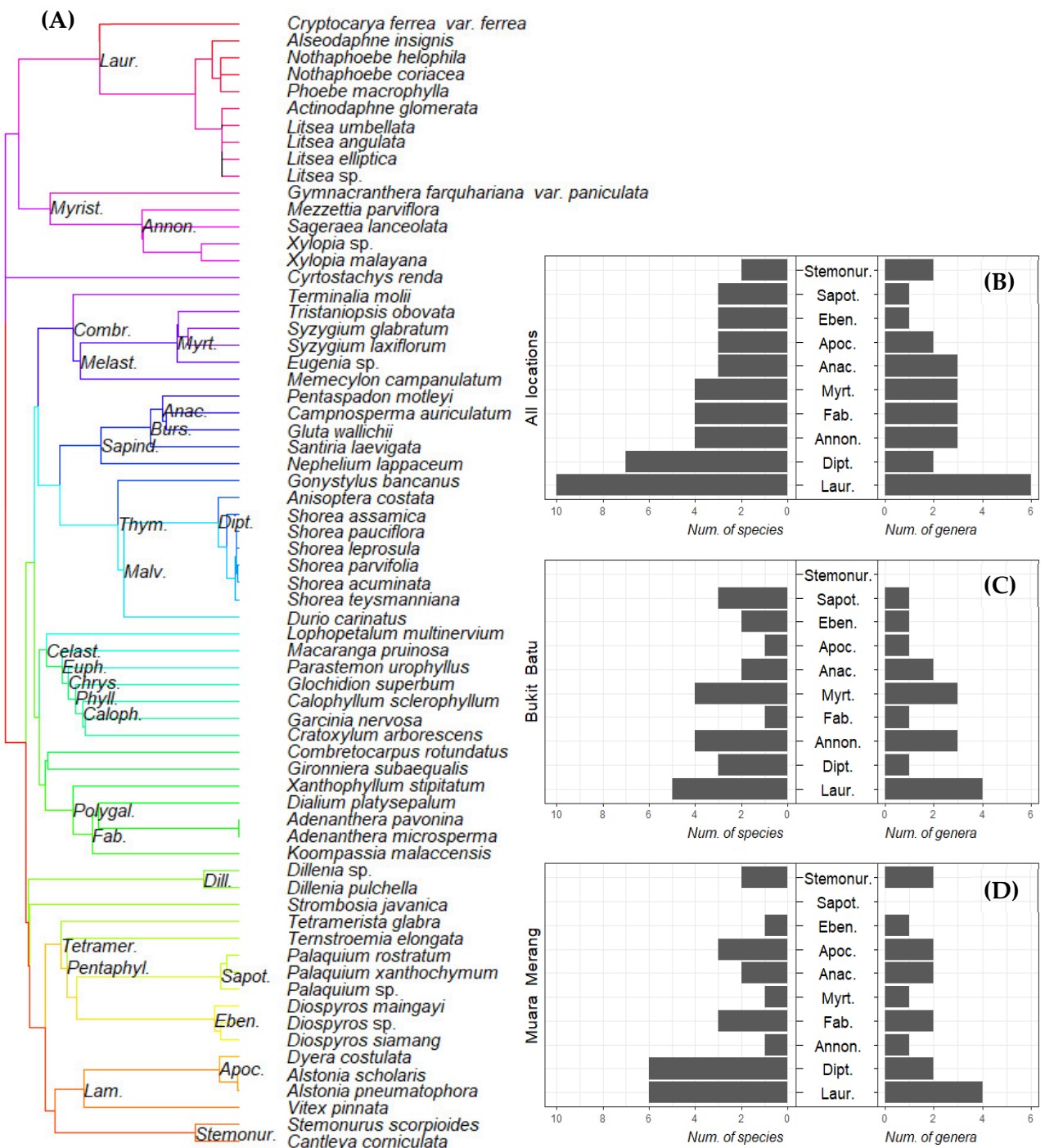

**Figure 3.** Species composition: (**A**) a phylogram of 67 species recorded across all locations and bar charts representing the number of species and genera from the ten largest families in (**B**) all locations, (**C**) Bukit Batu, and (**D**) Muara Merang.

**Table 1.** Dominant tree species with diameters ≥10 cm (IVI > 10%) in the research sites.

| No | Species | Local Name | Family | Density (N/ha) | IVI (%) |
|----|---------|------------|--------|----------------|---------|
| | | **Bukit Batu** | | | |
| 1 | *Palaquium xanthochymum* Pierre ex Burck | Pasir-pasir/Suntai | Sapotaceae | 116 | 66.27 |
| 2 | *Eugenia* sp. | Cemeti/Jambu-jambu | Myrtaceae | 62 | 32.76 |
| 3 | *Litsea* sp. | Medang | Lauraceae | 37 | 18.39 |
| 4 | *Xylopia malayana* Hook.f. & Thomson | Tempurung | Annonaceae | 35 | 17.22 |
| 5 | *Shorea teysmanniana* Dyer ex Brandis | Meranti bunga | Dipterocarpaceae | 23 | 14.62 |
| 6 | *Shorea pauciflora* King | Meranti sapat | Dipterocarpaceae | 26 | 14.15 |
| 7 | *Diospyros siamang* Bakh. | Arang-arang | Ebenaceae | 32 | 13.96 |
| 8 | *Gymnacranthera farquhariana* var. *paniculata* (A.DC.) R.T.A.Schouten | Dara-dara | Myristicaceae | 19 | 10.83 |
| 9 | *Garcinia nervosa* Miq. | Asam kandis | Clusiaceae | 20 | 10.42 |
| | | **Muara Merang** | | | |
| 1 | *Eugenia* sp. | Jambu-jambu | Myrtaceae | 156 | 60.88 |
| 2 | *Alseodaphne insignis* Gamble | Medang | Lauraceae | 58 | 26.34 |
| 3 | *Adenanthera pavonina* L. | Pisang-pisang/Saga | Fabaceae | 29 | 22.11 |
| 4 | *Dyera costulata* Hook.f. | Jelutung | Apocynaceae | 29 | 19.85 |
| 5 | *Shorea parvifolia* Dyer | Meranti | Dipterocarpaceae | 36 | 17.22 |
| 6 | *Cryptocarya ferrea* var. *ferrea* Blume | Medang pirangan/pergam | Lauraceae | 23 | 16.16 |
| 7 | *Diospyros* sp. | Arang-arang | Ebenaceae | 35 | 16.02 |
| 8 | *Koompassia malaccensis* Maingay | Manggris | Fabaceae | 15 | 12.07 |
| 9 | *Shorea pauciflora* King | Meranti sapat | Dipterocarpaceae | 20 | 10.42 |

### 3.1.2. Vegetation Species Potential

In the research sites in the Bukit Batu protected area, the species that may potentially replace future stands include sapling-level stands, such as pasir-pasir (*Palaquium xanthochymum* (de Vriese) Pierre.) (IVI = 23.51%), arang-arang (*Diospyros siamang* Bakh.) (IVI = 20.15%), and mensiro (*Mezzetia parviflora* (Hook.f. & Thomson) Oliv.) (IVI = 19.72%); these are presented in Table 2. In Muara Merang, they include jambu-jambu (*Eugenia* sp.) (IVI = 64.34%), kedondong hutan (*Pentaspadon motleyi* Hook.f.) (IVI = 26.97%), and medang (*Alseodaphne insignis* Gamble.) (IVI = 18.38%).

**Table 2.** Dominant saplings with IVI >10% in the research sites.

| No | Species | Local Name | Family | Density (N/ha) | IVI (%) |
|----|---------|------------|--------|----------------|---------|
| | | **Bukit Batu** | | | |
| 1 | *Palaquium xanthochymum* Pierre ex Burck | Pasir-pasir/Suntai | Sapotaceae | 19 | 23.51 |
| 2 | *Diospyros siamang* Bakh. | Arang-arang | Ebenaceae | 22 | 20.15 |
| 3 | *Mezzettia parviflora* Becc. | Mensiro | Annonaceae | 17 | 19.72 |
| 4 | *Garcinia nervosa* Miq. | Asam kandis | Clusiaceae | 14 | 14.47 |
| 5 | Litsea sp. | Medang | Lauraceae | 11 | 14.39 |
| 6 | *Ternstroemia elongata* (Korth.) Koord. | Nyamuk-nyamuk | Pentaphylacaceae | 12 | 14.13 |
| 7 | *Syzygium glabratum* (DC.) Veldkamp | Tulang-tulang | Myrtaceae | 9 | 11.46 |
| 8 | *Xanthophyllum stipitatum* A.W.Benn. | Kemuning | Polygalaceae | 9 | 10.59 |
| | | **Muara Merang** | | | |
| 1 | *Eugenia* sp. | Jambu-jambu | Myrtaceae | 32 | 64.34 |
| 2 | *Pentaspadon motleyi* Hook.f. | Kedondong hutan | Anacardiaceae | 12 | 26.97 |
| 3 | *Alseodaphne insignis* Gamble | Medang | Lauraceae | 7 | 18.38 |
| 4 | *Diospyros* sp. | Arang-arang | Ebenaceae | 6 | 12.73 |
| 5 | *Tetramerista glabra* Miq. | Punak | Tetrameristaceae | 5 | 11.62 |

In the research sites in the Bukit Batu protected area, the species that may potentially replace future stands include seedling-level stands such as pasir-pasir (*Palaquium xanthochymum* (de Vriese) Pierre.) (IVI = 51.67%), kelat (*Xylopia* sp.) (IVI = 31.06%), and mensiro (*Mezzetia parviflora* (Hook.f. & Thomson) Oliv.) (IVI = 16.33%); these are presented in Table 3. In Muara Merang, they include pisang-pisang/saga (*Adenanthera pavonina*)

(IVI = 66.41%), jambu-jambu (*Eugenia* sp.) (IVI = 25.41%), and medang (*Alseodaphne insignis* Gamble.) (IVI = 24.50%).

**Table 3.** Dominant seedlings with IVI >10% in the research sites.

| No | Species | Local Name | Family | Density (N/ha) | IVI (%) |
|---|---|---|---|---|---|
| | | **Bukit Batu** | | | |
| 1 | *Palaquium xanthochymum* Pierre ex Burck | Pasir-pasir/Suntai | Sapotaceae | 70 | 51.67 |
| 2 | *Xylopia* sp. | Kelat | Annonaceae | 26 | 31.06 |
| 3 | *Mezzettia parviflora* Becc. | Mensiro | Annonaceae | 16 | 16.33 |
| 4 | *Eugenia* sp. | Cemeti/Jambu-jambu | Myrtaceae | 17 | 15.44 |
| 5 | *Gymnacranthera farquhariana* var. *paniculata* (A.DC.) R.T.A.Schouten | Dara-dara | Myristicaceae | 10 | 10.56 |
| | | **Muara Merang** | | | |
| 1 | *Adenanthera pavonina* L. | Pisang-pisang/Saga | Fabaceae | 37 | 66.41 |
| 2 | *Eugenia* sp. | Jambu-jambu | Myrtaceae | 12 | 25.41 |
| 3 | *Alseodaphne insignis* Gamble | Medang | Lauraceae | 9 | 24.50 |
| 4 | *Alstonia scholaris* (L.) R.Br. | Pelai pipit | Apocynaceae | 10 | 16.83 |
| 5 | *Gluta wallichii* (Hook.f.) Ding Hou | Rengas burung | Anacardiaceae | 9 | 15.80 |

Table 4 presents an analysis that compares stand density and the tree species count, which serve as indicators of biodiversity richness in forest areas, with data from other peat regions.

**Table 4.** Comparison of stand density and number of tree species with diameters ≥10 cm between the research sites and with other locations from several sources.

| Location | Above Sea Level (m) | Plot Size (Ha) | Stand Density (N/ha) | Number of Species | Sources |
|---|---|---|---|---|---|
| Bukit Batu, Bengkalis, Riau | 22 | 1 | 528 | 38 | This study |
| Muara Merang, Banyuasin, SumSel | 20 | 1 | 565 | 39 | This study |
| Hutan Bukit Datuk, Dumai | 18.2 | 1 | 354 | 22 | [55] |
| Central Kalimantan | 0–50 | na | 1200–1825 | na | [24] |
| Selat Panjang, Riau | 0–50 | 1 | 550 | 49 | [27] |
| Teluk Meranti, Riau | 20 | 4 | - | 35 | [56] |
| Kampar, Riau | 12 | 1 | 478 | 27 | [57] |
| Senepis Peninsula | 10 | 0.2 | 535 | 25 | [58] |
| Bukit Batu Bengkalis, Riau | <15 | 0.1 | 530 | 19 | [59] |

*3.2. Stand Structure, Regeneration, and Conservation Status*

Figure 4 illustrates the overall stand structure of the trees in the research plot. In Bukit Batu, the tree species with dominant heights exceeding 25 m include pasir-pasir (*Palaquium xanthochymum* (de Vriese) Pierre.) at 32.6 m, meranti bunga (*Shorea teysmanniana* Dyer.) at 32.3 m, and pisang-pisang (*Diospyros maingayi* (Heiren) Bakh.) at 31.7 m; the species with dominant heights between 20 m and 25 m are jangkang (*Dillenia pulchella* (Jack) Gilg.) (24.7 m), meranti sapat (*Shorea gibbosa* Brandis.) (24.6 m), and meranti bunga (*Shorea teysmanniana* Dyer.) (24.4 m), while the species with heights of <20 m are jambu-jambu (*Eugenia* sp.) (19.9 m), milas (*Lophopetallum multinervium* Ridl.) (19.8 m), and pasir-pasir (*Palaquium xanthochymum* (de Vriese) Pierre.) (19.7 m). In Muara Merang, the tree species with dominant heights of >25 m are jelutong (*Dyera costulata* Hook.f.) (34.1 m), manggris (*Koompassia malaccensis* Maing.) (33.3 m), and medang pirangan (*Cryptocarya tomentosa* Blume.) (32.6 m); the species with dominant heights between 20 m and 25 m are pulai pipit (*Alstonia scholaris* (L) R. Br.) (24.4 m), jambu-jambu (*Eugenia* sp.) (23.8 m), and punak (*Tetramerista glabra* Miq.) (23.6 m), while the species with dominant heights of <20 m are medang hitam (*Alseodaphne insignis* Gamble.) (19.9 m), meranti merah (*Shorea parvifolia* Dyer.) (19.8 m), and manggris (*Koompassia malaccensis* Maing.) (19.8 m).

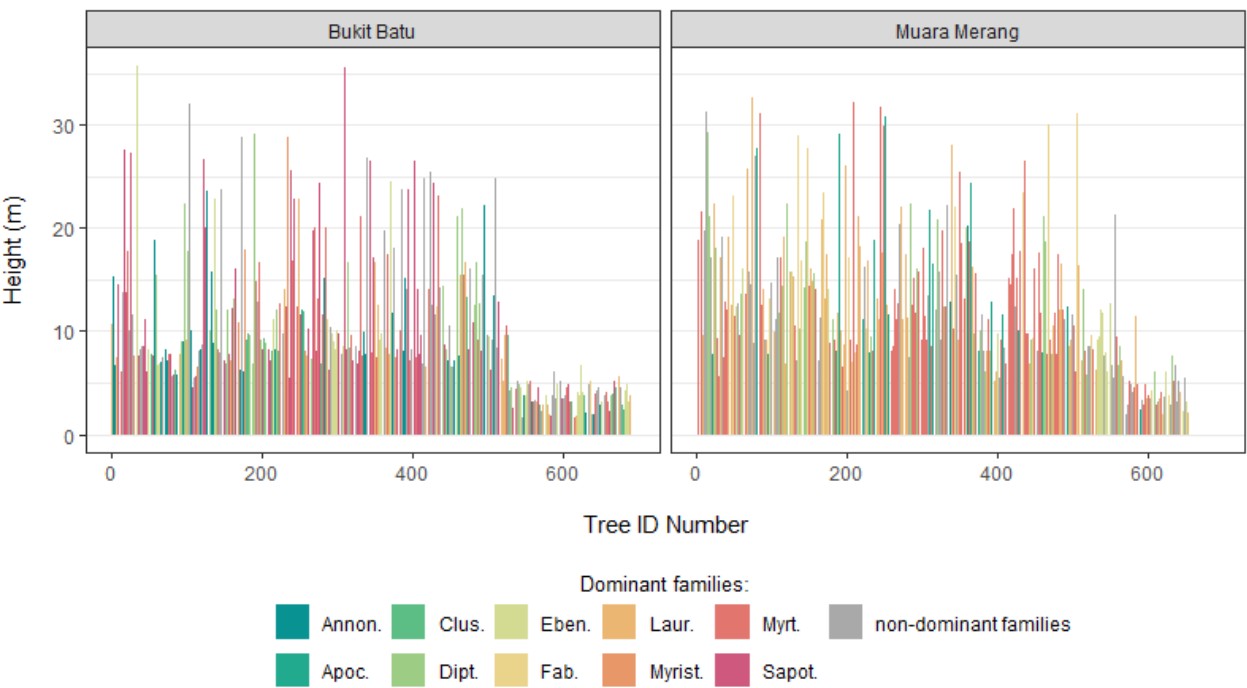

**Figure 4.** Stand profile of the forest at the research site.

The research findings depict the distribution of trees across the diameter classes spanning those from 10 to 19 cm, 20 to 29 cm, 30 to 39 cm, 40 to 49 cm, and those equal to or exceeding 50 cm in the research site, as shown in Figure 5. This bar plot highlights the reduction in the number of trees from the smaller to larger diameter classes and illustrates the stand structure in the secondary forests.

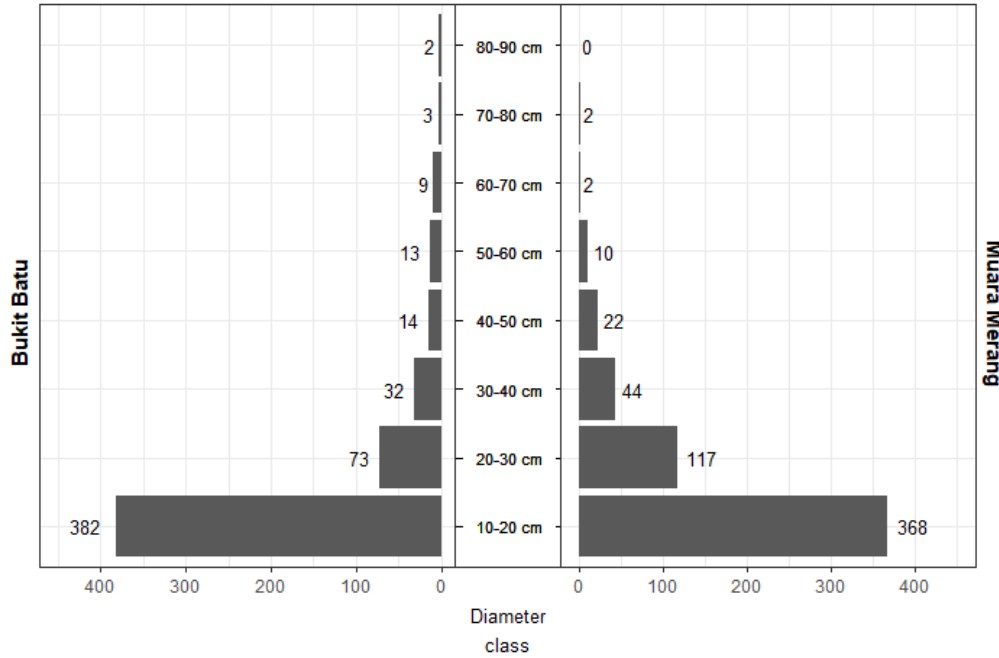

**Figure 5.** Stand structure of trees based on the relationship between diameter class and number of trees in the research site.

Figure 6 shows the regeneration of entire plant species, which are found across all of the stages (trees, saplings, and seedlings). Figure 6 shows that the dominant species in Bukit

Batu with complete regeneration from seedlings up to tree level is pasir-pasir (Palaquium xanthochymum (de Vriese) Pierre.) (IVI= 66.27%, 23.51%, and 51.67%). Meanwhile, in Muara Merang, it is jambu-jambu (Eugenia sp.) (IVI= 60.88%, 64.34%, and 25.41%).

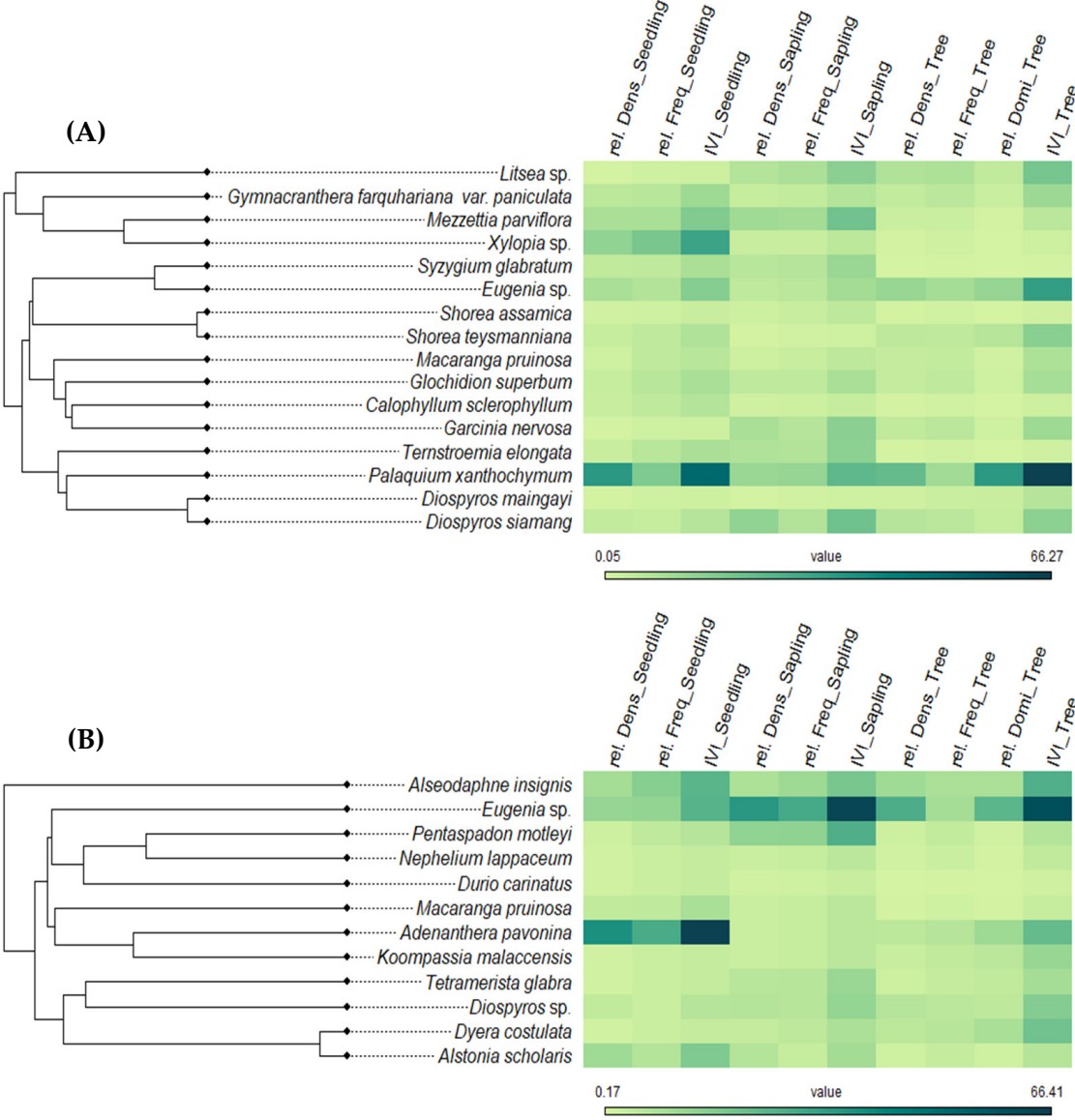

**Figure 6.** Tree species with complete regeneration across three vegetation growth stages: (**A**) Bukit Batu, Riau; (**B**) Muara Merang, South Sumatra.

Table 5 presents the documentation of 67 tree species in the Bukit Batu and Muara Merang research sites; their conservation status was evaluated following the guidelines from the International Union for Conservation of Nature's Red List of Threatened Species (IUCN), the Convention on International Trade in Endangered Species of Wild Fauna and Flora (CITES), and the relevant national regulations (P 7/1999 and P 106/2018).

**Table 5.** List of conservation status of tree species recorded in the research sites of Bukit Batu and Muara Merang.

| No | Botanical Name | Local Name | Family | Conservation Status | | | |
|---|---|---|---|---|---|---|---|
| | | | | Endemic | IUCN | CITES | P. 7/1999; P.106/MEN LHK/SETJEN/KUM.1/ 1/2/2018 |
| 1 | *Campnosperma auriculatum* Hook.f. | Terentang | Anacardiaceae | no | LC | - | - |
| 2 | *Gluta wallichii* (Hook.f.) Ding Hou | Rengas/Rengas burung | Anacardiaceae | no | LC | - | - |
| 3 | *Pentaspadon motleyi* Hook.f. | Kedongdong hutan | Anacardiaceae | no | DD | - | - |
| 4 | *Combretocarpus rotundatus* (Miq.) Danser | Prepat/Tertih | Anisophylleaceae | no | LC | - | - |
| 5 | *Mezzettia parviflora* Becc. | Mensiro | Annonaceae | no | NE | - | - |
| 6 | *Sageraea lanceolata* Miq. | Terpis | Annonaceae | no | LC | - | - |
| 7 | *Xylopia malayana* Hook.f. & Thomson | Tempurung | Annonaceae | no | NE | - | - |
| 8 | *Alstonia pneumatophora* Backer ex Den Berger | Pulai | Apocynaceae | no | LC | - | - |
| 9 | *Alstonia scholaris* (L.) R.Br. | Pelai pipit | Apocynaceae | no | LC | - | - |
| 10 | *Dyera costulata* Hook.f. | Jelutung | Apocynaceae | no | LC | - | - |
| 11 | *Cyrtostachys renda* Blume | Pinang merah | Arecaceae | no | NE | - | - |
| 12 | *Santiria laevigata* Blume | Para-para | Burseraceae | no | LC | - | - |
| 13 | *Calophyllum sclerophyllum* Vesque | Bintangur/Mentangur | Calophyllaceae | no | LC | - | - |
| 14 | *Lophopetalum multinervium* Ridl. | Milas | Celastraceae | no | NE | - | - |
| 15 | *Parastemon urophyllus* (Wall. ex A.DC.) A.DC. | Kelat milas | Chrysobalanaceae | no | NE | - | - |
| 16 | *Garcinia nervosa* Miq. | Asam kandis | Clusiaceae | no | NE | - | - |
| 17 | *Terminalia molii* Exell | Garam-garam | Combretaceae | yes | EN | - | - |
| 18 | *Dillenia pulchella* (Jack) Gilg | Jangkang | Dilleniaceae | no | NE | - | - |
| 19 | *Anisoptera costata* Korth. | Mersawa | Dipterocarpaceae | no | EN | - | - |
| 20 | *Shorea acuminata* Dyer | Meranti payau | Dipterocarpaceae | no | LC | - | - |
| 21 | *Shorea assamica* Dyer | Meranti putih | Dipterocarpaceae | no | LC | - | - |
| 22 | *Shorea leprosula* Miq. | Meranti rawa | Dipterocarpaceae | no | NT | - | - |
| 23 | *Shorea parvifolia* Dyer | Meranti | Dipterocarpaceae | no | LC | - | - |
| 24 | *Shorea pauciflora* King | Meranti sapat | Dipterocarpaceae | no | NT | - | - |
| 25 | *Shorea teysmanniana* Dyer ex Brandis | Meranti bunga | Dipterocarpaceae | no | CR | - | - |
| 26 | *Diospyros maingayi* (Hiern) Bakh. | Pisang-pisang | Ebenaceae | no | NE | - | - |
| 27 | *Diospyros siamang* Bakh. | Arang-arang | Ebenaceae | no | NE | - | - |
| 28 | *Diospyros* sp. | Arang-arang | Ebenaceae | | | - | - |
| 29 | *Macaranga pruinosa* Müll.Arg. | Mahang/Makaranga | Euphorbiaceae | no | NE | - | - |
| 30 | *Adenanthera microsperma* Teijsm. & Binn. | Sogo | Fabaceae | no | LC | - | - |
| 31 | *Adenanthera pavonina* L. | Pisang-pisang/Saga | Fabaceae | no | LC | - | - |
| 32 | *Dialium platysepalum* Baker | Asam keranji | Fabaceae | no | LC | - | - |

**Table 5.** *Cont.*

| No | Botanical Name | Local Name | Family | Conservation Status | | | |
|---|---|---|---|---|---|---|---|
| | | | | Endemic | IUCN | CITES | P. 7/1999; P.106/MEN LHK/SETJEN/KUM.1/ 1/2/2018 |
| 33 | *Koompassia malaccensis* Maingay | Manggris | Fabaceae | no | LC | - | - |
| 34 | *Cratoxylum arborescens* (Vahl) Blume | Geronggang | Hypericaceae | no | LC | - | - |
| 35 | *Vitex pinnata* L. | Laban | Lamiaceae | no | LC | - | - |
| 36 | *Actinodaphne glomerata* (Blume) Nees | Medang lendir | Lauraceae | no | LC | - | - |
| 37 | *Alseodaphne insignis* Gamble | Medang | Lauraceae | no | LC | - | - |
| 38 | *Litsea angulata* Blume | Medang lendir/putih | Lauraceae | no | NE | - | - |
| 39 | *Litsea elliptica* Blume | Medang perawas | Lauraceae | no | LC | - | - |
| 40 | *Litsea* sp. | Medang | Lauraceae | | | - | - |
| 41 | *Litsea umbellata* (Lour.) Merr. | Medang kirangan | Lauraceae | no | LC | - | - |
| 42 | *Durio carinatus* Mast. | Durian hutan | Malvaceae | no | NT | - | - |
| 43 | *Memecylon campanulatum* C.B.Clarke | Pelam-pelam | Melastomataceae | no | NE | - | - |
| 44 | *Syzygium glabratum* (DC.) Veldkamp | Tulang-tulang | Myrtaceae | no | NE | - | - |
| 45 | *Syzygium laxiflorum* DC. | Tenggek burung | Myrtaceae | yes | NT | - | - |
| 46 | *Tristaniopsis obovata* (Benn.) Peter G.Wilson & J.T.Waterh. | Pelawan | Myrtaceae | no | NE | - | - |
| 47 | *Strombosia javanica* Blume | Kacang-kacang | Olacaceae | no | NE | - | - |
| 48 | *Ternstroemia elongata* (Korth.) Koord. | Nyamuk-nyamuk | Pentaphylacaceae | yes | NE | - | - |
| 49 | *Glochidion superbum* Baill. ex Müll.Arg. | Samak | Phyllanthaceae | no | NE | - | - |
| 50 | *Xanthophyllum stipitatum* A.W.Benn. | Kemuning | Polygalaceae | no | NE | - | - |
| 51 | *Nephelium lappaceum* L. | Rambutan hutan | Sapindaceae | no | LC | - | - |
| 52 | *Palaquium rostratum* (Miq.) Burck | Suntai putih | Sapotaceae | no | LC | - | - |
| 53 | *Palaquium xanthochymum* Pierre ex Burck | Pasir-pasir/Suntai | Sapotaceae | no | VU | - | - |
| 54 | *Cantleya corniculata* (Becc.) R.A.Howard | Dara-dara | Stemonuraceae | no | VU | - | - |
| 55 | *Stemonurus scorpioides* Becc. | Sebencik | Stemonuraceae | no | NE | - | - |
| 56 | *Tetramerista glabra* Miq. | Punak | Tetrameristaceae | no | VU | - | - |
| 57 | *Gonystylus bancanus* (Miq.) Kurz | Ramin | Thymelaeaceae | no | CR | II | - |

Note: NE = not evaluated, DD = data deficient, LC = least concern, NT = near threatened, VU = vulnerable, EN = endangered, CR = critically endangered; II = Appendix II.

*3.3. Biomass and Carbon Stock*

Using the equation proposed by Chave et al. [48], Table 6 presents the biomass and carbon stock of the forest stands with diameters equal to or exceeding 10 cm in the research site.

**Table 6.** The estimated above-ground biomass, carbon stock, and sequestered carbon dioxide equivalent in Bukit Batu and Muara Merang old secondary forests.

| Diameter Class (cm) | Estimation Based on Chave's Equation | | |
| --- | --- | --- | --- |
| | Biomass (Tonnes/ha) | Carbon Stock (Tonnes C/ha) | CO$_2$ Sequestration (Tonnes CO$_2$eq/ha) |
| **Bukit Batu research site** | | | |
| 10–19 | 25.25 | 11.89 | 43.56 |
| 20–29 | 22.04 | 10.36 | 38.02 |
| 30–39 | 24.10 | 11.33 | 41.58 |
| 40–49 | 22.09 | 10.38 | 38.10 |
| ≥50 | 84.61 | 39.77 | 145.94 |
| Total | 178.10 | 83.70 | 307.20 |
| **Muara Merang research site** | | | |
| 10–19 | 29.93 | 14.07 | 51.63 |
| 20–29 | 38.45 | 18.07 | 66.33 |
| 30–39 | 39.36 | 18.50 | 67.89 |
| 40–49 | 38.39 | 18.04 | 66.22 |
| ≥50 | 44.27 | 20.81 | 76.37 |
| Total | 190.41 | 89.49 | 328.44 |

## 4. Discussion

The findings of this research indicate that Bukit Batu has more species and families than Muara Merang; however, the latter has more trees with a diameter of ≥10 cm. This will obviously influence the amount of biomass in both locations. According to Istomo and Farida [60], biomass originates from living vegetation, including stems, tree crowns, undergrowth, weeds, and annual plants, which have the capacity to absorb carbon from the atmosphere through photosynthesis. Lutz et al. [61] also stated that large trees produce a higher amount of above-ground biomass. As a result, large-diameter trees have a significant impact on forest biomass.

The dominant tree species in Bukit Batu include *Palaquium xanthochymum*, *Eugenia* sp., and *Litsea* sp., and in Muara Merang, the dominant tree species are *Eugenia* sp., *Alseodaphne insignis*, and *Adenanthera pavonine* (Table 1). The research by Sadili [62] in the Giam Siak Kecil and Peat Forest of PT Arara Abadi stated that the dominant species were *Campnosperma squamatum*, *Mangifera parvifolia*, *Mezzettia havilandii*, and *Gonystylus bancanus*. These results are in agreement with those of a study in Southern Peninsula, Malaysia—which was treated as one landscape—in which the dominant species included *Syzygium cenirum*, *Syzygium inoplyllum*, *Palaquium hexandruma*, *Stemonorus sequndiflorus,* and *Koompassia mallacensis* [63]. The findings suggest that economically vital species, such as *Gonystylus bancanus*, exhibit limited occurrence, while key peatland species, such as Meranti bunga (Shorea teysmanniana) and Meranti sapat (*Shorea gibbosa*) from the Dipterocarpaceae family and Arang-arang (Diospyros siamang) are present. Subsequently, in the Muara Merang research areas, the key species of peatland that included Dipterocarpaceae were recorded.

In undisturbed forests (natural forests) in Sebangau, more than one hundred species were found [64]. Furthermore, in the same location, Denny and Kalima [65] found fewer species: a maximum of 96; thus, the species composition appears to be highly dynamic, with variations significantly influenced by location and environmental conditions, which shape both the composition and the structural aspects of the stand. The total number of tree species near the large and small channel block observation of Sebangau National Park [66] is higher than in Bukit Batu and Muara Merang. The variation in the number of species shows that the peat swamp forest species composition is highly dynamic and is related to

the location and time of observation [67]. Another influencing factor is changes in land use. These changes cause variations in the composition of the species that constitute the stand [64].

Generally, because of degradation and agricultural conversion, peatland forests in Sumatra were defined as secondary forests [12,68]. The Bukit Batu and Muara Merang (Table 4) protected areas have a moderate-to-high density of stand and species numbers as these forest areas are old secondary forests in relatively good condition [54].

The forest stand structure with a decreasing number of trees in classes with diameters ranging from small to large formed an inverted "J" curve shape (Figure 5). The forest's stand structure in the research area exhibited normal growth characteristics. Generally, smaller diameter classes are more commonly found than larger diameter classes in natural forests [69]. Young trees will naturally replace older ones following logging, fire, falling (natural disasters), or physiological death. This process is known as regeneration. Stand structure analysis provides important information on the spatial organization and distribution patterns of plants in forest ecosystems and takes into account both vertical and horizontal dimensions. In the vertical dimension, this research assessed species distribution across canopy levels and considered factors such as height stratification, canopy closure, and species composition. Horizontally, it evaluated tree distribution across diameter classes, providing data on stand density, age structure, and growth dynamics [36,55]. This comprehensive approach to stand structure study is critical for understanding ecosystem dynamics and devising effective forest management strategies. Pretzsch et al. [70] emphasized the need to take into account both vertical and horizontal dimensions when studying stand structure in order to better understand forest ecosystems and to enhance management practices.

One of the factors determining the C content in ecosystems is the basal area. The variation in the basal area in the two locations affected the biomass and C content in the vegetation. In the natural environment, the water table is at or near the peat surface throughout most of the year. This condition rises with rainfall and falls due to outflow from the surface and evapotranspiration [71]. However, disturbances such as logging, deforestation, and fire in and around the peat swamp forest can reduce organic matter supply and peat formation, which has an impact on ecosystem biodiversity. Many trees show adaptations to waterlogged or nutrient-poor environments as a defense strategy [72].

The stand structure in both locations shows that the ratio of the number of trees with a diameter of 10–20 cm to the other diameters in Bukit Batu is relatively different from that in Muara Merang. The number of trees in the 20–30 cm diameter class in Bukit Batu is only 19.1% of the number of 10–20 cm class trees, while in Muara Merang, it reaches 31.8%. This may indicate differences in the disturbances that occur in the tree vegetation in the two locations. Even though rare fires can reduce the species richness and diversity of tropical peat swamp forests, Volkova et al. [19] found that two to three decades after fire, levels of richness and diversity that are similar to those of relatively undisturbed reference forests are reached. Given the normal growth represented by the reversed 'J' curve in the stand structure of the secondary peat forest in our study area (Figure 5), future ecosystem regeneration in both study sites is promising. A regeneration process is reflected by complete profiles of individuals along with a gradient of diameters from seedlings to trees with the largest diameter [73].

Regeneration is crucial in the formation of a forest. Achard et al. [74] and Hansen and DeFries [75] mentioned that the regrowth of secondary vegetation on previously degraded and cleared land is the second most significant change in land use in the tropics. In our study, since the stand structure indicates regeneration in the secondary forest, we can predict that Bukit Batu and Muara Merang areas will play a significant role in the natural restoration of peatland forest ecosystems in Sumatra.

Concerning carbon sequestration, particularly in tropical climates, natural forest regeneration is often regarded as an efficient low-cost method for sequestering carbon [76,77]. Worldwide estimates indicate that 24% to 35% of all carbon emissions from fossil fuels,

and industrial production from 2000 to 2010 could be mitigated if tropical deforestation were completely stopped, mature forests remained undisturbed, and new forests were allowed to continue regrowing on deforested land [78]. In the management context of our research location, forests that function as protected areas have an additional impact on the improvement in biodiversity conservation and the sustainability of carbon sequestration in these areas.

Regarding conservation status, the species *Shorea teysmanniana*, commonly known as meranti bunga, has a critically endangered status according to the IUCN Red List [79]. Seven species are classified as least concern (LC), one as near threatened (NT), two as vulnerable (VU), and one as data deficient (DD). The remaining 17 species have not yet been evaluated. None of the 29 species are listed under the CITES or included in national regulations (P 7/1999 and P 106/2018) (Table 5). For meranti bunga in particular, the CR status indicates that the species is at an extremely high risk of extinction in the wild.

While the regeneration process in tropical peat swamp forests shows promising signs of recovery after fires, it is essential to address the conservation status of endangered species such as *Shorea teysmanniana*, or meranti bunga, which faces extreme risks due to habitat loss, degradation, fragmentation, and overexploitation, and to highlight the importance of implementing effective conservation efforts to prevent extinction. For *S. teysmaniana* in particular, the CE status indicates that the species is at an extremely high risk of extinction in the wild. Contributing factors to its endangered status may include habitat loss, degradation, fragmentation, and the overexploitation of timber and other resources. Therefore, conservation efforts are crucial to prevent its extinction.

In terms of conservation priority, endemic and critically endangered species stand out as they are most susceptible to extinction. These species inhabit niches with precise environmental conditions and typically have limited numbers, genetic diversity, and geographic range. They also face challenges such as low reproductive rates, weak competitive abilities, and reliance on dispersal [80–82]. These factors make them highly vulnerable to human activities and various other threats. Therefore, it is essential to promptly put into action effective conservation strategies to protect these species before they disappear completely.

Conservation strategies encompass both in situ and ex situ programs. In situ conservation involves protecting organisms and ecosystems in their natural habitats, while ex situ conservation involves preserving species outside of their native environments. Critically endangered species such as *S. teysmaniana* require conservation actions both within and outside their natural habitats. In situ efforts focus on preventing damage to species and habitats, facilitating natural propagation. Ex situ conservation for these species involves using various technologies to support their expansion. Similarly, vulnerable species require in situ endeavors to safeguard both their existences and their habitats, accompanied by ex situ strategies comparable to those employed for critically endangered species. Near-threatened species derive advantages from precautionary actions such as those that manage land use alterations while taking their survival into account.

In the protected regions of Bukit Batu, forest stands with a diameter of 10 cm or more possess a moderate biomass and carbon stock (Table 6). This equates to 178.10 tonnes biomass/ha or 83.70 tonnes C/ha, which is equivalent to 307.20 tonnes $CO_2$/ha. In contrast, in Muara Merang, the figure was 190.41 tonnes biomass/ha, or 89.49 tonnes C/ha, which is equivalent to 328.44 tonnes $CO_2$/ha. Our C stock carbon findings were lower than the C stock at a 479–564 cm depth of degraded peat swamp forest in Tumbang Nusa, Central Kalimantan, which was 95.2 tonnes C/ha [6]. Moreover, they were notably lower than the quantities observed in the secondary peat swamp forest located in Katingan Regency, Central Kalimantan, which totaled 1752.4 ± 401.3 Mg-C/ha [24]. This difference is due to the contrast in the analytical approaches used; Saragi-Sasmito et al. [24] conducted a thorough evaluation of every aspect of the carbon cycle, encompassing carbon storage, emissions, and sequestration, in both above-ground and below-ground carbon reservoirs within the research area. However, our results indicate higher carbon values than those of the primary peat forests in Central Kalimantan, which was 73.08 tonnes C/ha [44].

Forest biomass is expressed as dry weight per unit area, which comprises foliage, flowers, fruits, branches, small branches, main stems, roots, and fallen or deceased trees [27]. The amount of forest biomass is determined by measuring diameter, height, wood density, plant density, and soil fertility [25,27,83]. Estimating biomass in tropical forests is highly necessary as it influences the carbon cycle [55] and thus draws the attention of all related stakeholders to the need to conserve the carbon stock. This is true since approximately 47% of the peat swamp forests' biomass is carbon [52]. The significant results of this observation are in line with those reported by Chave et al. [48] and Natalia et al. [84].

Young trees within forest stands show great potential to reduce atmospheric carbon dioxide levels due to their rapid growth compared to older trees. During the growth and photosynthesis process, carbon dioxide and water are converted into carbohydrates and are subsequently metabolized into lipids, nucleic acids, and proteins, and eventually transformed into various plant organs [85]. The above-ground productivity of peat swamp forests plays a crucial role in the process of carbon sequestration in ecosystems [24].

## 5. Conclusions

This study recorded 42 and 36 plant species in the protected areas of Bukit Batu and Muara Merang, respectively, on plots covering 1 ha; the species were classified into 26 and 20 families, respectively. Trees with a diameter of ≥10 cm comprised 38 species and a total of 528 trees/ha in Bukit Batu, whereas Muara Merang had 565 trees/ha from 39 species. The dominant tree species in Bukit Batu included *Palaquium xanthochymum*, *Eugenia* sp., and *Litsea* sp., with important value indexes of 66.27%, 32.76%, and 18.39%, respectively. In Muara Merang, the dominant tree species were *Eugenia* sp., *Alseodaphne insignis*, and *Adenanthera pavonina*, with important value indexes of 60.88%, 26.34%, and 22.11%, respectively. The stand diversity indexes (H') in Bukit Batu and *Muara Merang* were classified as medium, with values of 2.93 and 2.82. The dominant species of the tree, sapling, and seedling stages in Bukit Batu was *Palaquium xanthochymum,* with important value indexes of 66.27%, 23.51%, and 51.67%, respectively. Meanwhile, in Muara Merang, the dominant species of the tree, sapling, and seedling stages was *Eugenia* sp., with important value indexes of 60.88%, 64.34%, and 25.41%, respectively. The forest stand structure in the research site showed a decreasing number of trees that ranged in diameter from small to large classes, generating an inverted "J" curve shape. This revealed that the forest stand structure in the research site had normal growth characteristics. The forest stands in Bukit Batu with a diameter of > 10 cm had a biomass and carbon stock of 178.10 tonnes/ha, or 83.70 tonnes C/ha, which is equal to 307.20 tonnes $CO_2$/ha. Meanwhile, in Muara Merang, it was 190.41 tonnes/ha, or 89.49 tonnes C/ha, which is equal to 328.44 tonnes $CO_2$/ha.

This research in Bukit Batu, Riau Province, enriches the existing data and supports the restoration efforts in the Giam Siak Kecil Biosphere Reserve (designated by UNESCO's Man and Biosphere Program) in Riau Province. Our research underscores the necessity of enhancing vegetation structure, composition, and carbon stock to represent biodiversity and stand growth effectively. Therefore, conserving the remaining old secondary peat swamp forests, particularly forests within the research study areas that are designated as 'protected areas', is crucial to preserve biodiversity and maintain their ecological function as carbon pools and regulators in tropical ecosystems.

**Author Contributions:** All authors have equal roles as main contributors in this study. I.W.S.D. performed the conceptual ideas and the outline, conducted the literature reviews, performed the analysis, prepared the initial draft, provided critical feedback on each section, and revised and finalized the manuscript. N.M.H. conducted the literature reviews, performed the analysis, provided critical feedback on each section, and revised and finalized the manuscript. R.G. conducted the literature reviews, performed the data interpretation, provided critical feedback on each section, and revised and finalized the manuscript. R.T.K. conducted the literature reviews, performed the data interpretation, provided critical feedback on each section, and revised and finalized the manuscript. R.S. conducted the literature reviews, performed the data interpretation, provided critical feedback on each section, and revised and finalized the manuscript. D. conducted the literature reviews, prepared

the initial draft, provided critical feedback on each section, and revised and finalized the manuscript. T.S. conducted the literature reviews, performed the analysis, provided critical feedback on each section, and revised and finalized the manuscript. P. conducted the literature reviews, performed the data interpretation, provided critical feedback on each section, and revised and finalized the manuscript. B.H.N. conducted the literature reviews, performed the analysis, provided critical feedback on each section, and revised and finalized the manuscript. C.A.S. conducted the literature reviews, performed the data interpretation, provided critical feedback on each section, and revised and finalized the manuscript. I.K.A. performed the analysis, performed the data interpretation, provided critical feedback on each section, and revised and finalized the manuscript. All authors have read and agreed to the published version of the manuscript.

**Funding:** This research was funded by the Forest Research & Development Center with funding number 134/P3h/Proev/Lit.0/2.

**Data Availability Statement:** The datasets used/and/or analyzed during the current study are available through FigShare: Dharmawan et al. (2024) Vegetation dynamics [LAND] at https://dx.doi.org/10.6084/m9.figshare.25661784 (accessed on 22 April 2024).

**Acknowledgments:** We thank the anonymous reviewers for their detailed comments and corrections.

**Conflicts of Interest:** The authors declare no conflicts of interest.

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
