# Peer review of "The Dynamics of Vegetation Structure, Composition and Carbon Stock in Peatland Ecosystem of Old Secondary Forest in Riau and South Sumatra Provinces"

_land, doi:10.3390/land13050663_

Round 1
Reviewer 1 Report
Comments and Suggestions for Authors
General comment
The manuscript presents vegetation biodiversity and carbon stock assessment data from two peat swamp forest sites in Sumatra. The dataset will be important to support the national greenhouse gas inventory program in Indonesia. The manuscript is well-prepared, but I have my major concern regarding data availability from this work. I would recommend authors make the data available through a public repository, such as figshare for transparency and future usability. I also noticed that the formatting needs to be improved. Please find below some of my specific comments.
Specific comment
Line 3: I would suggest dropping the word “potency” in the title
Line 28: it should be carbon stocks or carbon density instead of carbon content
Line 29: “similiar” do you mean ‘equal to’ instead?
Line 44: I would suggest not using etc in scientific publication.
Line 47-48: please reword these two sentences (a bit hard to understand), also it is unclear how peatlands have a connection with peat swamp forests here. Some peatlands have no forest cover.
Line 56-58: ‘renowned’, do you mean important? Suggest to reword this sentence
Line 60: for me, it is hard to find the connection between high biomass and food security
Line 65: Please apply subscript for CO2 here and throughout the manuscript
Line 70-71: no vegetation found, where? All over Indonesia’s peatlands? Suggest to reword this sentence
Line 75: BRGM terms this as Peat Hydrology Unit instead of Area, no?
Line 81: Please use ‘carbon stocks or carbon density’ instead of ‘carbon content’, carbon content usually indicates % of carbon for a particular sample.
Line 94-95: I think the readers need a bit more details information on the study aims, objectives and how your findings can fill the described gaps. Make them a new paragraph if needed.
Line 119: am I correct that one location has one 100 x 100m plot? I would also suggest that the sampling date should be mentioned.
Line 134: please provide a bit more information about the figure in the caption.
Line 135: no statistical analysis?
Line 184: GREAT figures! Well done! Can you add the label of x axis for Fig 3 B,C and D?
Line 209: I think you can find additional data from Ref 26 representing Kalimantan region
Line 263: carbon stocks!
Sigit
James Cook University
Comments on the Quality of English LanguageNo comment
Author Response
Dear Reviewer 1
Thank you very much for your appreciations and suggestions. All revisions have been made in the manuscript as suggested.
This manuscript language also has been edited by Language Author Service MDPI.
Best regards,
I Wayan S. Dharmawan
Reviewer No. |
Comments |
Responses |
1 |
The manuscript is wellprepared, but I have my major concern regarding data availability from this work. I would recommend authors make the data available through a public repository, such as figshare for transparency and future usability.
|
Thank you very much for your appreciations and suggestions. We have made the data available through Figshare as per the reviewer’s suggestion (line 502-504)
|
|
Line 3: I would suggest dropping the word “potency” in the title
Line 28: it should be carbon stocks or carbon density instead of carbon content
Line 29: “similiar” do you mean ‘equal to’ instead?
Line 44: I would suggest not using etc in scientific publication.
|
It has been revised in line 3
It has been revised in line 19, 29, 183-185, 281284, 420-433, 470
It has been revised in line 30
It has been revised in line 45
|
|
Line 47-48: please reword these two sentences (a bit hard to understand), also it is unclear how peatlands have a connection with peat swamp forests here. Some peatlands have no forest cover.
Line 56-58: ‘renowned’, do you mean important? Suggest to reword this sentence
Line 60: for me, it is hard to find the connection between high biomass and food security
Line 65: Please apply subscript for CO2 here and throughout the manuscript
|
In fact, most of the peatlands exist as peat swamp forest landscapes and are perceived as fragile ecosystems. Any disturbances to the vegetation can have an adverse effect on the peat’s characteristics and its hydrological function. (line 50-52)
In relation to the current global climate change issue, the C pool in the peatlands of Indonesia is of great significance as they accumulate and conserve as much as 17-19% (65 Gt) of the global peat C pool [7]. (line 61-64)
Lowland tropical rainforests, including peat swamp forests, typically exhibit high biodiversity and relatively high biomass [11]. (Line 65-66) Note: delete references [12,13].
From 2000 to 2016, the carbon dioxide (CO2) emissions from the forestry sector (logging and fires) averaged 0.71 Gt CO2-e [15,16]. (Line 73-74). Note: the subscript CO2 has been applied through out the manuscript.
|
|
Line 70-71: no vegetation found, where? All over Indonesia’s peatlands? Suggest to reword this sentence
|
Following the 2015 peatland fire in Indonesia, limited vegetation that was suitable for seedlings and other plants was observed, except for a small number of fern species that exhibited the highest density [20]. (Line 79-81)
|
|
Line 75: BRGM terms this as Peat Hydrology Unit instead of Area, no?
Line 81: Please use ‘carbon stocks or carbon density’ instead of ‘carbon content’, carbon content usually indicates % of carbon for a particular sample.
Line 94-95: I think the readers need a bit more details information on the study aims, objectives and how your findings can fill the described gaps. Make them a new paragraph if needed.
|
The landscape is managed by Kawasan Hidrologi Gambut (KHG) or the Peat Hydrology Unit [22]. (Line 84-85)
Studies of the carbon stock conditions of several peat forests in Indonesia, showed a carbon stock of 73.08 tons C/ha in the burned forests with diameters ≥10 cm in Central Kalimantan's peat forests [25] (Line 90-92). Note: carbon stock has been applied through out the manuscript.
This study aims to assess vegetation structure, composition, and carbon stock potential in protected areas by focusing on secondary peat forests in Sumatra, Indonesia. By addressing the knowledge gaps, we aim to support conservation and sustainable management efforts and contribute to biodiversity preservation and climate change mitigation at the regional and global levels. (Line 105-109)
|
|
Line 119: am I correct that one location has one 100 x 100m plot? I would also suggest that the sampling date should be mentioned.
Line 134: please provide a bit more information about the figure in the caption.
Line 135: no statistical analysis?
|
Yes, that one location has one 100 x 100m plot. The research study was conducted from May 2022 to August 2022. (Line 136-138)
The title in Fig. 2 has been changed to be “Research measurement plot in Bukit Batu and Muara Merang.” (Line 154)
Yes, there is no statistical analysis in this study.
|
|
Line 184: GREAT figures! Well done! Can you add the label of x axis for Fig 3 B,C and D?
|
We have revised the figures (Line 204-208) |
|
Line 209: I think you can find additional data from Ref 26 representing Kalimantan region
Line 263: carbon stocks!
|
The information has been added in line 232-234
It has been revised in line 281. Note: carbon stock has been applied through out the manuscript. |

Reviewer 2 Report
Comments and Suggestions for Authors
The manuscript entitled “The Dynamics of Vegetation Structure-Composition and Carbon Stock Potency at Peatland Ecosystem of Old Secondary Forest in Riau and South Sumatra Provinces” by Wayan Susi Dharmawan and collaborators is a relevant contribution to understanding the dynamic of tropical forests that deserves to be published. The information in this work is helpful to be considered for restoration and revegetation practices when environmental impacts occur. However, several changes must be made before publication. I annotated some comments and suggestions in the manuscript.
In a general overview, the work is structured correctly and well-written. There are some weak points in the manuscript that I will highlight below.
Methods: It is not clear enough how you calculated the important value index (IVI) and which equation was used. You probably can fix it by providing the equation and adding more detail about this calculation.
Discussion: It is focused on the published results and conclusions of other works. A more robust discussion of your results is expected.
Conclusions: The findings of this study should be highlighted in this section. I strongly recommend to avoid repeat the results.
I hope my comments can be useful for improving the work. I am sure that the authors can provide a nice revised work if they focus their effort on improving each highlighted point in the text.

In a general overview, the work is structured correctly and well-written.
Author Response
Dear Reviewer 2
Thank you very much for your appreciations and suggestions. All revisions have been made in the manuscript as suggested.
This manuscript language also has been edited by Language Author Service MDPI.
Best regards,
I Wayan S. Dharmawan
Reviewer No. |
Comments |
Responses |
2 |
The manuscript entitled “The Dynamics of Vegetation Structure-Composition and Carbon Stock Potency at Peatland Ecosystem of Old Secondary Forest in Riau and South Sumatra Provinces” by Wayan Susi Dharmawan and collaborators is a relevant contribution to understanding the dynamic of tropical forests that deserves to be published. |
Thank you very much for your appreciations and suggestions. All detailed suggestions/inputs have been incorporated in throughout the manuscript. |
|
The information in this work is helpful to be considered for restoration and revegetation practices when environmental impacts occur. However, several changes must be made before publication. I annotated some comments and suggestions in the manuscript. |
|
|
Methods: It is not clear enough how you calculated the important value index (IVI) and which equation was used. You probably can fix it by providing the equation and adding more detail about this calculation. |
Thank you for the valuable comment. We have added an equation for calculating IVI along with the explanations (line 163-169) |
|
Discussion: It is focused on the published results and conclusions of other works. A more robust discussion of your results is expected.
|
We have added more robust discussion as suggested. The total number of tree species near the large and small channel block observation of Sebangau National Park [66] is higher than in Bukit Batu and Muara Merang. The variation in the number of species shows that the peat swamp forest species composition is highly dynamic and is related to the location and time of observation [67]. (Line 314-318) The stand structure in both locations shows that the ratio of the number of trees with a diameter of 1020 cm to the other diameters in Bukit Batu is relatively different from that in Muara Merang. The number of trees in the 20-30 cm diameter class in Bukit Batu is only 19.1% of the number of 10-20 cm class trees, while in Muara Merang it reaches 31.8%. This may indicate differences in the disturbances that occur in the tree vegetation in the two locations. (Line 351-356) Regeneration is crucial in the formation of a forest. Achard et al. [74] and Hansen and DeFries [75] mention that the regrowth of secondary vegetation on previously degraded and cleared land is the second most significant change in land use in the tropics. In our study, since the stand structure indicates a regeneration in the secondary forest, we can predict that the Bukit Batu and Muara Merang areas will play a significant role in the natural restoration of peatland forest ecosystems in Sumatra. (Line 364-369) Concerning carbon sequestration, particularly in tropical climates, natural forest regeneration is often regarded as an efficient low-cost method for sequestering carbon [76,77]. Worldwide estimates indicate that 24 to 35% of all carbon emissions from fossil fuels and industrial production from 2000 to 2010 could be mitigated if tropical deforestation were completely stopped, mature forests remained undisturbed, and new forests were allowed to continue regrowing on deforested land [78]. In the management context of our research location, the forests that function as protected areas have an additional impact on the improvement of biodiversity conservation and the sustainability of carbon sequestration in the areas. (Line 370-378) |
|
|
While the regeneration process in tropical peat swamp forests shows promising signs of recovery after fires, it is essential to address the conservation status of endangered species such as Shorea teysmanniana, or meranti bunga, which faces extreme risks due to habitat loss, degradation, fragmentation, and overexploitation, and to highlight the importance of implementing effective conservation efforts to prevent extinction. This species has a Critically Endangered status according to the IUCN Red List [79]. Seven out of twenty-nine species are classified as Least Concern (LC), one as Near Threatened (NT), two as Vulnerable (VU), and one as Data Deficient (DD), The remaining 17 species have not yet been evaluated. None of these species are listed under CITES or included in national regulations (P 7/1999 and P 106/2018). For S. teysmaniana in particular, CE status indicates that the species is at an extremely high risk of extinction in the wild. Contributing factors to its endangered status may include habitat loss, degradation, fragmentation, and overexploitation of timber and other resources. Therefore, conservation efforts are crucial to prevent its extinction. (Line 386-399) In terms of conservation priority, endemic and critically endangered species stand out as they are most susceptible to extinction. These species inhabit niches with precise environmental conditions and typically have limited numbers, genetic diversity, and geographic range. They also face challenges such as low reproductive rates, weak competitive abilities, and reliance on dispersal [80–82]. These factors make them highly vulnerable to human activities and various other threats. Therefore, it is essential to promptly put into action effective conservation strategies to protect these species before they disappear completely. (Line 400-407) Conservation strategies encompass both in-situ and ex-situ programs. In-situ conservation involves protecting organisms and ecosystems in their natural habitats, while ex-situ conservation involves preserving species outside of their native environments. Critically endangered species such as S. teysmaniana requires conservation actions both within and outside their natural habitats. In-situ efforts focus on preventing damage to species and habitats, facilitating natural propagation. Ex-situ conservation for these species involves using various technologies to support their expansion. Similarly, vulnerable species require in-situ endeavors to safeguard both their existences and their habitats, accompanied by ex-situ strategies comparable to those employed for critically endangered species. Near threatened species derive advantages from precautionary actions such as those which manage land use alterations while taking their survival into account. (Line 408-419) Estimating biomass in tropical forests is highly necessary as it influences the carbon cycle [55] and thus draws the attention of all related stakeholders to the need to conserve the carbon stock. This is true since approximately 47% of the peat swamp forests’ biomass is carbon [52]. The significant results of this observation are in line with those reported by Chave et al. [48] and Natalia et al. [84]. [75]. (Line 438-443) |
|
Conclusions: The findings of this study should be highlighted in this section. I strongly recommend to avoid repeat the results. |
It has been revised in line 456-472 |

Reviewer 3 Report
Comments and Suggestions for Authors
This is a unique and valuable paper on peatland ecosystems.
If you have any additional information on the following points, please add additional explanations.
This study reports on vegetation structure, and trees can be broadly divided into sun trees and shade trees. Additionally, their photosynthetic and metabolic abilities are different. Furthermore, the physiological effects of ectomycorrhizal and endomycorrhizal tree species are different. Please consider the differences in carbon content between these sun and shade trees, as well as between ectomycorrhizal and endomycorrhizal tree species.
Author Response
Dear Reviewer 3
Thank you very much for your appreciations and suggestions. All revisions have been made in the manuscript as suggested.
This manuscript language also has been edited by Language Author Service MDPI.
Best regards,
I Wayan S. Dharmawan
Reviewer No. |
Comments |
Responses |
3 |
This is a unique and valuable paper on peatland ecosystems. If you have any additional information on the following points, please add additional explanations. This study reports on vegetation structure, and trees can be broadly divided into sun trees and shade trees. Additionally, their photosynthetic and metabolic abilities are different. Furthermore, the physiological effects of ectomycorrhizal and endomycorrhizal tree species are different. Please consider the differences in carbon content between these sun and shade trees, as well as between ectomycorrhizal and endomycorrhizal tree species. |
Thank you very much for your appreciations and suggestions.
So far, we do not have yet the observation in this study on the effect of several tree layers or mycorrhizal merits on the carbon biomass |
